# A register-based study of long-term health and social care costs among children with prenatal alcohol exposure

**Mirjami Jolma**[1,2]*, **Mikko Koivu-Jolma**[3,4], **Taisto Sarkola**[5,6], **Mika Gissler**[7,8,9], **Niina-Maria Nissinen**[10], **Hanna Kahila**[11], **Anne Sarajuuri**[12], **Paulus Torkki**[1], **Ilona Autti-Rämö**[1,12], **Anne Koponen**[13,14]

**1** Faculty of Medicine, The University of Helsinki, Helsinki, Finland, **2** Division of Child Neurology, Päijät-Häme Central Hospital, Lahti, Finland, **3** Faculty of Science, The University of Helsinki, Helsinki, Finland, **4** Department of Environmental and Biological Sciences, University of Eastern Finland, Joensuu, Finland, **5** New Children's Hospital, University of Helsinki and Helsinki University Hospital, Helsinki, Finland, **6** Minerva Foundation Institute for Medical Research, Helsinki, Finland, **7** Department of Data and Analytics, THL Finnish Institute for Health and Welfare, Helsinki, Finland, **8** Department of Molecular Medicine and Surgery, Karolinska Institute and Academic Primary Health Care Centre, Region Stockholm, Stockholm, Sweden, **9** Research Centre for Child Psychiatry, University of Turku, Turku, Finland, **10** Faculty of Social Sciences, Health Sciences Unit, Tampere University, Tampere, Finland, **11** Department of Obstetrics and Gynecology, University of Helsinki and Helsinki University Hospital, Helsinki, Finland, **12** Division of Child Neurology, University of Helsinki, New Children's hospital, Helsinki, Finland, **13** Folkhälsan Research Center, Public Health Research Program, Helsinki, Finland, **14** Department of Public Health, University of Helsinki, Helsinki, Finland

* mirjami.jolma@helsinki.fi

## Abstract

Prenatal alcohol exposure (PAE) associated with fetal alcohol spectrum disorders (FASD) often remain underdiagnosed. They globally cause a wide range of health and social problems leading to high costs. To outline cumulative health and social care costs in children related with PAE with and without diagnosed FASD, we followed 427 children with PAE until the age of 20 years, and 1795 controls born 1992–2001 until the year 2016. All hospital care and out-of-home care episodes, including placements in foster or residential care, were analyzed, and their costs estimated. Age-dependent patterns of diagnoses and costs of those with PAE with and without diagnosed FASD were compared to controls. Children with PAE had significantly higher risks and hospital costs for both somatic and psychiatric conditions compared with controls. Mean cumulative hospital costs were 55500€ (IQR, interquartile range, 56800€) for PAE with FASD, 30100€ (IQR 25100€) for PAE without FASD and 15600€ (IQR 12000€) for controls. Between 0–10 years, FASD was associated with higher somatic costs, whereas psychiatric costs dominated in the PAE without FASD group. FASD diagnosis was associated with lower risks of traumatic injuries, substance use disorders, and teenage pregnancies, independent of early out-of-home care, which was associated with FASD. Out-of-home care was common in PAE groups, and its costs far exceeded hospital costs: mean cumulative costs were

**Data availability statement:** The data that support the findings of this study are available from the Hospital District of Helsinki and Uusimaa (HUS). However, restrictions apply to the availability of these sensitive data, which were used under license for the current study. Only those researchers who are named in the study permissions have access to the data, and no part of the data can be shared or placed in public repositories. Similar data can be applied from Findata Finnish Social and Health Data Permit Authority Findata https://findata.fi/en/ Link to the current legislation: https://findata.fi/en/services-and-instructions/legislation/.

**Funding:** MJ received Open Access funding from Helsinki University Library for the publication of this article. The authors received no additional funding for this work.

**Competing interests:** The authors have declared that no competing interests exist.

30-fold in FASD (610000€, IQR 375800€) and 17-fold in others with PAE (344300€, IQR 621900€) compared to controls (20500€, IQR 0€). Health and particularly social care costs associated with PAE are significant. High out-of-home care costs reflect substantial need for support for families at risk. Early diagnosis of FASD may mitigate secondary complications and associated costs emerging in adolescence. Prevention policies are urgently needed at primary, secondary and tertiary level.

## Introduction

Estimated global prevalence of alcohol use in pregnancy is 9.8% [1]. This equals over 13 million children at risk for problems associated with prenatal alcohol exposure (PAE) among the annually born 130 million children [2]. In Europe 2% of children and youth are estimated to have fetal alcohol spectrum disorders (FASD) [3], however even much higher prevalence rates in some areas of the world have been reported such as 31% in Western Cape, South Africa [4]. FASD includes lifelong neurodevelopmental, psychiatric, cognitive, and physical disorders such as congenital anomalies and stunted growth caused by PAE [5,6]. Over 400 somatic and psychiatric clinical manifestations and comorbidities have been associated with FASD [7,8] and heavy PAE [9] with higher prevalence than in the general population.

In studies on the effects of PAE, the main problem is that only a small minority of those with FASD are currently diagnosed [10]. Persistent underdiagnosis underestimates and hides the etiological cause for challenges in a significant part of the population with performance difficulties due to PAE and renders research on the societal consequences of PAE to be biased.

Cost estimations of FASD have varied due to differences in populations, time periods, societal systems, and methods [11]. Most studies have been conducted in the USA and Canada [11] where social and health care systems differ from European models. Estimations typically have included only those with dysmorphic fetal alcohol syndrome (FAS) [11], the severe type of FASD that is easiest to diagnose due to facial features and growth disorder in addition to neurobehavioral disorders present in all types of FASD [5]. European studies on cost estimations of FASD are still scarce. In Sweden, which is also a Nordic welfare state with tax-funded universal healthcare and child protection services, similar to neighboring Finland, mean annual societal cost for a child with FAS was estimated to be 76 000€, totaling 1.37 million euros by age of 18 years corresponding to total societal cost of €1600 million per year using a conservative estimated prevalence 0.2% [12]. In comparison, in Portugal with similar population size to Sweden, total annual societal costs of prevalent childhood asthma were estimated as €161 million [13]. In Finland, our clinical study of children with FASD estimated total costs per child at about €1 million by age 16, including costs for in- and outpatient care, medications, rehabilitations, out-of-home care, disability allowance, and special education arrangements [14]. Underdiagnosis of FASD causes underestimation of the full economic impact of PAE and a bias in cost estimations as only those on the severe end of the FASD spectrum have been diagnosed and controls are likely to include a significant number of undiagnosed cases.

Costs of FASD reflect long-term needs for services in social welfare, health care, education, justice system, financial support, and loss of parental and individual income, some of which are easier to estimate than others [11,12]. Children with FASD are overrepresented in out-of-home care, including foster and residential care [15], which is the last resort of child protection services. In Finland, major costs associated with FASD in children have been shown to be caused by out-of-home care [14]. Health care costs are mainly caused by inpatient hospital care, emergency, and outpatient clinic visits [14]. These two types of costs, out-of-home and hospital care can be estimated using comprehensive national register data that enables long-term retrospective follow-up of both cases and controls with minimal drop-out rate caused only by death or emigration.

In Finland it is recommended in prenatal care to ask about alcohol and drug use before and during pregnancy from all pregnant women, and if continuing use during pregnancy is revealed, women will be referred to intensified pregnancy follow-up in tertiary health care. Previous register-based research of children prenatally exposed to maternal substance use in Finland has shown high risk for out-of-home care [16–18], which has been associated with mental and behavioral disorders [19], and an increased number of somatic diagnoses and more frequent use of medications [16–18]. The previous articles based on the data on children prenatally exposed to maternal substance use have included both alcohol and drug exposed children as an aggregated group. As alcohol is both a widely consumed legal substance and a well-recognized teratogen, the fetal effects and postnatal social effects of maternal alcohol use disorders may differ from other types of maternal substance use disorders. The evidence of alcohol teratogenicity and fetal toxicity is robust and based on both human research and animal models spanning over 50 years [e.g., 6,20–26], whereas the research on teratogenic effects of psychoactive drugs has still been limited or yielded inconclusive results [27–31]. Compared to PAE and FASD, much less is known for possible long-term health and developmental effects of prenatal drug exposures. Thus, the costs of alcohol exposed cases among prenatally substance exposed children need to be analyzed separately.

Costs associated with PAE with and without FASD or related socioeconomic long-term effects of maternal alcohol use disorder on offspring have seldom been accounted for in alcohol policy decisions due to lack of data. The costs of those diagnosed with FAS have been studied much more than the costs of other types of FASD and there has been little research on costs associated with PAE without diagnosed FASD. More research on the whole spectrum is clearly needed to estimate the true costs due to PAE considering its high prevalence worldwide.

In this register-based long-term retrospective follow-up study of PAE children, we evaluated direct costs for hospital care and out-of-home care in relation to typical diagnoses seen in PAE. The main goal was to identify and quantify sources of health and social care costs of PAE children with or without a diagnosis of FASD.

## Research aims

- to identify diagnoses in PAE children with and without FASD diagnosis compared with control children without known PAE

- to estimate direct costs of in- and outpatient hospital care in PAE children with and without FASD diagnosis compared with controls until the age of 20 years, and

- to estimate direct costs of out-of-home care in PAE children compared with controls until the age of 20 years.

## Data and methods

Several previous articles concerning the outcome in children and youth with prenatal substance exposure based on the same data have been published as a part of larger project ADEF Helsinki – alcohol or drug exposure during fetal life. Publications have included themes such as healthcare utilization and welfare interventions [16–18], maternal welfare and morbidity [32], adverse childhood experiences [19,33,34], neurodevelopmental [33,34], mental and behavioral [19,35], and

socioeconomic problems in youth [36]. The cohort and the registers used have been presented in detail in a published cohort profile [37].

The original long-term follow up case cohort consists of 622 children born 1.1.1992–21.4.2001 and their mothers who had been followed-up and treated during pregnancy in a specialized maternity clinic for mothers with substance use disorder. The data of cases included information about maternally disclosed alcohol and drug use during and before pregnancy, together with the information about whether the pregnancy was planned and information concerning the time of the first prenatal care visit. The non-exposed control cohort consists of 1798 child and mother dyads without evidence of maternal substance misuse in national health and social welfare registers during one year before childbirth (2420 mother and child dyads altogether). Three mother–child control dyads were matched with each exposed mother–child dyad for maternal age, parity, number of fetuses, month of delivery, and delivery hospital of the index child.

Register records were available until 31.12.2016. Consequently, the records were available until the age of 15–24 years depending on the birth year of each child. Health care records included all child inpatient hospital care episodes. Outpatient hospital care data were available from 1.1.1998 with outpatient hospital care data missing for children born 1.1.1992–31.12.1997. Medical birth register data were available from 1.1.1992 but detailed neonatal diagnoses were available only from 1.1.1996. For all age-specific cost estimations, only individuals with available data for the age were included. For the present study, researchers accessed the data set for the first time 6.6.2021.

From the original 622 cases, whose mothers had been referred for intensified evaluation and follow-up in a specialized tertiary clinic during pregnancy due to suspected or documented harmful alcohol and/or drug use during pregnancy, we excluded cases with probable drug exposure only and included those whose mothers met any of the following criteria:

- admitted alcohol use during pregnancy

- admitted daily alcohol use before pregnancy and admitted that pregnancy was unplanned

- admitted weekly alcohol use before pregnancy and the first prenatal health care visit was after gestational age of 12 weeks, or

- had been treated for any alcohol-related problems during the 9 months before birth.

In all, 427 cases were classified in the PAE group of this study. FASD-diagnosed cases were selected separately from child health care records based on any of the following: ICD-10 diagnosis Q86.0 (FAS or partial FAS), or at birth an ICD-10 diagnosis P04.3 (Fetus and newborn affected by maternal use of alcohol), or who before the ICD-10 era had ICD-9 code 7607A (Alcohol affecting fetus or newborn via placenta or breast milk) including FAS or Fetal alcohol effects (FAE, which is an older term for other types of FASD than FAS), or had an entry in the national Register of Congenital Malformations for FAS or FAE or suspected FAS/FAE or alcohol-related birth defect including listed features of microcephaly, flat philtrum, small palpebral fissures and thin upper lip. Among PAE cases, 48 children with FASD were detected. Three controls were discarded due to diagnosis P04.3 in the birth register or typical features of FAS listed in the Register of congenital malformations.

Thus, the final analyses included 427 PAE cases including 48 with FASD diagnosis and 379 without, and 1795 controls without known PAE or FASD diagnosis. Full follow-up records were available until 15 years for all cases and controls. Full out-of-home care follow-up records and health records until the age of 18 years were available for 269 (63.0%) of those with PAE (34 with FASD and 235 without) and for 1064 (59.3%) of the controls. Full health follow-up records until the age of 20 years were available for 158 (37%) of those with PAE (26 with FASD and 132 without) and for 602 (33.5%) of controls. Only those with full follow-up were included in analyses covering ages from 15 to 20 years.

Smoking and unplanned pregnancies were typical for mothers in both PAE groups (Table 1). Mothers of PAE with FASD children had disclosed daily alcohol use more frequently than mothers of PAE without FASD children. For controls only the information concerning smoking during pregnancy was available: 351 (19.6%) of controls had admitted smoking, significantly less than in PAE groups (p < 0.001). There were 104 controls where the mothers had alcohol-related diagnosis

**Table 1. Maternal self-reported prenatal exposure -related factors in PAE with and without FASD diagnosis.**

| Category | PAE with FASD n = 48 Freq (%) | PAE without FASD n = 379 Freq (%) | RR | 95% confidence intervals | p-value |
|---|---|---|---|---|---|
| Daily alcohol use before recognition of pregnancy | 33 (68.8) | 122 (32.2) | 2.1 | 1.7–2.7 | <0.0001 * |
| Unplanned pregnancy | 35 (72.9) | 270 (71.2) | 1.0 | 0.9–1.2 | 0.8249 |
| First prenatal visit after first trimester of pregnancy | 27 (56.2) | 129 (34) | 1.7 | 1.2–2.2 | 0.0035 * |
| Daily alcohol use during pregnancy | 16 (33.3) | 50 (13.2) | 2.5 | 1.6–4.1 | 0.0010 * |
| Weekly alcohol use during pregnancy | 17 (35.4) | 128 (33.8) | 1.0 | 0.7–1.6 | 0.8137 |
| Unplanned pregnancy and daily alcohol use | 15 (31.2) | 40 (10.6) | 3.0 | 1.8–4.9 | 0.0004 * |
| Unplanned pregnancy and weekly alcohol use | 13 (27.1) | 92 (24.3) | 1.1 | 0.7–1.8 | 0.6620 |
| Smoking during pregnancy | 43 (89.6) | 349 (92.1) | 1.0 | 0.9–1.1 | 0.5420 |
| Psychoactive drug use during pregnancy | 1 (2.1) | 41 (10.8) | 0.2 | 0.0–1.4 | 0.0404 * |

RR signifies risk ratio, calculated with *riskratio* function from *R* package *Epitools*, using Wald's method and two-sided exact mid-P method for p-values. Statistically significant p-values are marked with a star.

Psychoactive drug use signifies using opioids, benzodiazepines, cannabinoids, amphetamine or other psychoactive drugs.

within their lifetimes, though none of them had such diagnosis during one year before birth, and 43 (41.3%) of them had admitted smoking during pregnancy.

Children with PAE were significantly smaller at birth compared with controls, and those diagnosed with FASD were smaller than others with PAE and had shorter gestational lengths compared both with controls and others with PAE, all p < 0.0001. (Table 1.5 in S1 Appendix)

## Diagnoses and hospital in- and outpatient care costs

We analyzed all inpatient and outpatient hospital care episodes and extracted all registered diagnoses using three-digit ICD-10 codes (letter and two numbers). ICD-9 codes were mapped to ICD-10. We identified the 50 most common diagnoses in each group, FASD (n = 48), PAE without FASD (n = 379) and controls (n = 1795), excluding Z- codes, which primarily reflected family or social context rather than the child's health.

We calculated the prevalences of these most common diagnoses in the two PAE groups and compared each to controls. Diagnoses significantly more common in either PAE group than in controls (RR ≥ 1.5, 95% CI ≥ 1, p < 0.05) were identified (Tables 1.3a and 1.3b in S1 Appendix). These were then aggregated into diagnostic categories representing hallmark health and developmental challenges in PAE. One such category included diagnoses typically indicating a need for special educational support, such as intellectual disabilities (F70–F79), developmental disorders (F80–F89), and behavioral/emotional disorders of childhood (F90–F98).

For diagnosis-related costs we used diagnosis-related-group (DRG) prices extracted from publicly available Finnish health care unit price lists published in 2001, 2006, 2011 and 2017 [38–41] (see S1 Appendix). Nominal prices for other years were extrapolated using health care price index. In Finnish public healthcare the hospitals are non-profit, and the price levels should reflect the actual costs for the hospitals. For diagnoses without DRG price the medical specialty of treatment was determined using the ICD-10 group main diagnosis belonged to (see S1 Appendix) and:

- inpatient care cost was calculated according to the length of hospital stay multiplied by the average fee for a hospital day for each specialty, and

- outpatient care cost was calculated according to specialty clinic fee.

As a minor number of earlier outpatient clinic visits did not include a diagnosis, all undiagnosed outpatient clinic visits were classified for cost estimations as a general pediatric clinic visit for children under 16 years and as general medicine visits for those 16 years and over.

All costs were analyzed for complete age in years. For cumulative cost estimates, we analyzed costs for each age until the age of 20 years as the means of individual cumulative costs. We also analyzed psychiatric and somatic visits and costs separately. For index corrections we used the method described earlier by Jolma et al. (2023) [14]. Information about the annual cost per person was completed only until the age of 16 years due to the lack of information for a significant number of persons after that age. We reported the mean, median and interquartile range (IQR) for the costs.

## Out-of-home care and its costs

The age at the first out-of-home care placement was used for creating a Kaplan-Meier curve for proportion surviving without any out-of-home care comparing PAE with FASD, PAE without FASD, and non-PAE controls. To find out if out-of-home care would protect from hospitalizations, we first tested the association between out-of-home care and presence of hospital care. We also tested the association between FASD diagnosis and presence of hospital care. We treated these as categorical variable associations. Following this, we tested the effect of the first out-of-home care episode at the age of three or less on the number of hospitalizations. Finally, we tested for the overall effect of the age at first out-of-home care episode and hospitalizations. In the last two cases, we treated the response as an ordinal variable, while the observation variables in the first two tests were categorical, and continuous in the last two tests, respectively.

The unit costs per day for out-of-home care were extracted from the estimation report on child protection service costs for years 2005–2006 in the six biggest cities in Finland [42] (see S2 Appendix) together with Finnish social care unit price lists, first published in 2007, and then in 2011 and 2017 [40,41,43] (see S2 Appendix). The first source was selected as most of the cases and controls lived in three of these cities and no national unit price list for social care prices before 2007 was available, and the first national price list in 2007 was also based on it. The different types of out-of-home care, including foster family care, group homes and institutional care, have separate fees (see S2 Appendix).

The information on out-of-home care was typically available only for the age at first placement, list of different placement types provided separately for time periods 1992–2001 and 2002–2016, and total number of days in all types of out-of-home care combined. Consequently, the number of days in each care type (family, group home, or institution) had to be estimated. To this end, we divided the total number of placement days by the number of placement types during each time period. For those with longer institutional placements in each period it underestimates real costs and for those with short institutional placements it overestimates the costs. However, the majority had only one type of placement for each time period.

The index corrections were performed as explained for hospital costs estimating first nominal expenses for each year and then index correcting all costs to year 2023 level. The costs of out-of-home care in PAE with and without FASD diagnosis were compared with controls and total hospital and out-of-home care aggregate costs were estimated for each group for ages 0–20 years.

## Statistical analysis

All analyses were performed using R Statistical Software (v4.4.2) [44]. Risk ratios for categorical associations between groups were compared using *riskratio* function from the R package Epitools [45]. Because the out-of-home care age distributions and all cost distributions were extremely skewed, to test the differences between the groups we used, as recommended by Karch (2021) [46], the robust and nonparametric Brunner-Munzel's test [47,48]. The test was implemented in R package lawstat (v3.6) [49,50]. To compensate for multiple testing, we reported all test results with the exact p-values down to <0.0001.

## Ethical declaration

The study was approved by the Hospital district of Helsinki and Uusimaa (research permit HUS400/2016). The authorities maintaining the registers approved the use of data for this study project. The register linkages and pseudonymization of

the data were conducted by the Finnish Institute of Health and Welfare (THL) as the statistical authority. Therefore, the researchers did not access the registers directly and could not identify individual subjects. Because 1) study subjects were not contacted, 2) the study was considered to be of public health importance, 3) the number of participants was large, and 4) the research was done using official permission, according to the Finnish legislation (the Medical Research Act (488/1999)) the research was not considered medical research. Following this, separate statement from ethics committee or informed consent from the subjects were not required.

## Results

### Diagnoses

Typical diagnoses differed significantly in the three groups: PAE with FASD, PAE without FASD diagnosis, and controls. From the 50 most common diagnoses in the PAE groups, those significantly different from controls were identified (Table 1.3 in S1 Appendix), and aggregated by ICD-10 chapters and blocks (Table 2). Developmental, behavioral and emotional

**Table 2. Aggregated diagnose groups (ICD-10 diagnose chapters and blocks) of diagnoses differing from controls in PAE with and without FASD.**

| Aggregated diagnose groups | FASD (F) | PAE no FASD (P) | CONTROLS (C) | RR | 95% confidence intervals | P-value |
|---|---|---|---|---|---|---|
| | N = 48 | N = 379 | N = 1795 | F-C = FASD-CTRL | F-C = FASD-CTRL | F-C = FASD-CTRL |
| | n (%) | n (%) | n (%) | P-C = PAE-CTRL | P-C = PAE-CTRL | P-C = PAE-CTRL |
| | | | | | | |
| Congenital anomalies Q00-Q89 except Q86.0 (FAS) | 38 (79.2) | 56 (14.8) | 235 (13.1) | F-C 6.0 | F-C 5.0–7.3 | F-C < 0.0001 * |
| | | | | P-C 1.1 | P-C 0.8–1.6 | P-C = 0.3816 |
| Eyesight disorders H50-H59 | 18 (37.5) | 28 (7.4) | 83 (4.6) | F-C 8.1 | F-C 5.3–12.4 | F-C < 0.0001 * |
| | | | | P-C 1.6 | P-C 1.0–2.6 | P-C = 0.0335 * |
| Low birth weight P05, P07 | 13 (27.1) | 37 (9.8) | 128 (7.1) | F-C 3.8 | F-C 2.3–6.2 | F-C < 0.0001 * |
| | | | | P-C 1.4 | P-C 0.9–2.1 | P-C = 0.0866 |
| Developmental, behavioral and emotional disorders F70-F98 | 28 (58.3) | 149 (39.4) | 339 (18.9) | F-C 3.1 | F-C 2.4–4.0 | F-C < 0.0001 * |
| | | | | P-C 2.1 | P-C 2.2–3.5 | P-C < 0.0001 * |
| --Developmental disorders F70-F89 | 17 (35.4) | 65 (17.2) | 200 (11.1) | F-C 3.2 | F-C 2.1–4.8 | F-C < 0.0001 * |
| | | | | P-C 1.5 | P-C 1.2–2.3 | P-C = 0.0018 * |
| --Behavioral and emotional disorders F90-F98 | 25 (52.1) | 131 (34.6) | 224 (12.5) | F-C 4.2 | F-C 3.1–5.6 | F-C < 0.0001 * |
| | | | | P-C 2.8 | P-C 2.8–4.8 | P-C < 0.0001 * |
| Mood and anxiety disorders F30-F48 | 14 (29.2) | 107 (28.2) | 229 (12.8) | F-C 2.3 | F-C 1.4–3.6 | F-C = 0.0032 * |
| | | | | P-C 2.2 | P-C 2–3.5 | P-C < 0.0001 * |
| Respiratory tract diseases J09-J47 | 31 (64.6) | 180 (47.5) | 715 (39.8) | F-C 1.6 | F-C 1.3–2.0 | F-C = 0.0007 * |
| | | | | P-C 1.2 | P-C 1.1–1.7 | P-C = 0.0062 * |
| Alcohol or drug use disorder F10-F19 | 2 (4.2) | 37 (9.8) | 33 (1.8) | F-C 2.3 | F-C 0.6–9.2 | F-C = 0.2914 |
| | | | | P-C 5.3 | P-C 3.5–9.7 | P-C < 0.0001 * |
| Pregnancy, childbirth and the puerperium O00-O99 < 19 years | 0 (0) | 20 (5.3) | 36 (2.0) | F-C 0.0 | F-C - | F-C = 0.3832 |
| | | | | P-C 2.6 | P-C 1.5–4.9 | P-C = 0.0009 * |
| Traumatic injuries S00-S99, T00-T35 | 17 (35.2) | 176 (46.4) | 685 (38.2) | F-C 0.9 | F-C 0.6–1.4 | F-C = 0.7098 |
| | | | | P-C 1.2 | P-C 1.1–1.8 | P-C = 0.0030 * |

RR signifies risk ratio, calculated with *riskratio* function from *R* package *Epitools*, using Wald's method and two-sided exact mid-P method for p-values. Statistically significant p-values are marked with a star.

disorders of childhood, which often indicate a need for school support, were common in both PAE groups. In contrast, alcohol and drug use disorders, teenage pregnancies, and traumatic injuries were significantly more frequent only in the PAE without FASD group. These differences were not explained by variations in out-of-home care or age at first placement. Congenital anomalies (ICD-10 Q00-Q89) were unexpectedly frequent even among controls, with ankyloglossia (Q38.1) in 31 individuals (1.7%) being their most common anomaly (see Table 1.4 in S1 Appendix for details). Both PAE groups displayed significantly higher rates of hospital care for psychiatric diagnoses compared to controls (Fig 1)

## Costs of hospital care

Somatic health care costs were highest in infancy and among those diagnosed with FASD (Table 3, Fig 2). While psychiatric costs increased with age in all groups, the increase was especially steep in PAE without FASD (Fig 3c and 3d).

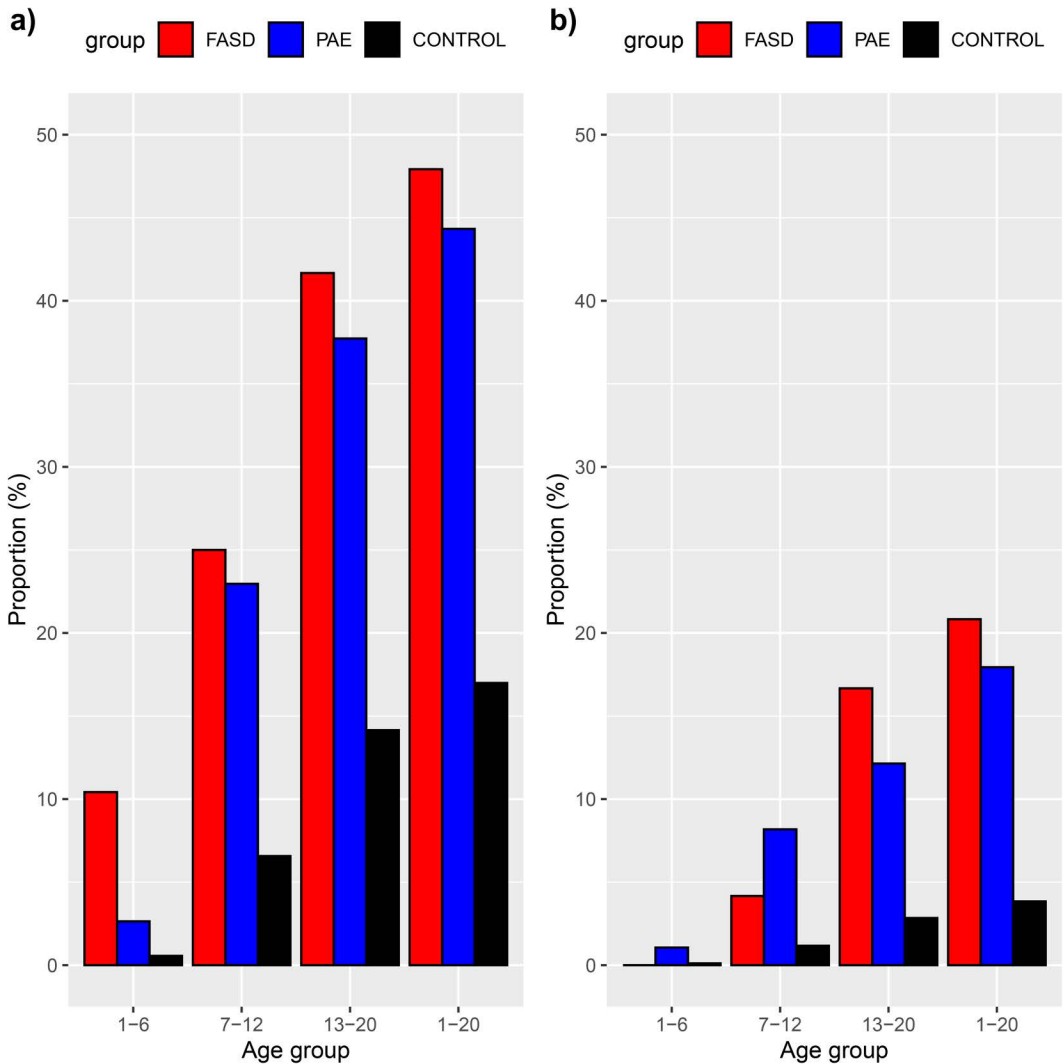

**Fig 1. Psychiatric care prevalences in different age groups (FASD in red, PAE without FASD in blue and Controls in black), a) for outpatient psychiatric care and b) for inpatient psychiatric care.** Numbers for the age categories 1–6, 7–12 and 13–20 years include only the age specific prevalence, whereas the category 1–20 describes the cumulative prevalence for the whole measured period. Psychiatric diagnoses included ICD-10 diagnoses: F10-F69 and F84-F98.

**Table 3. Annual estimated costs of all hospital care per person in different age categories rounded to full euros, and the proportion of persons with hospital care, of those with full information in each group.**

| Age category | FASD | PAE no FASD | Controls |
|---|---|---|---|
| < 1 year Mean € | 20824 | 5840 | 4819 |
| < 1 year Median (IQR) € | 9499 (8889) | 2765 (5730) | 532 (4111) |
| < 1 year Proportion having hospital care/of those with full information available (%) | 22/22 (100) | 174/215 (80.9) | 573/923 (62.1) |
| 1-6 years Mean/year € | 2535 | 700 | 452 |
| 1-6 years Median (IQR)/year € | 1495 (2893) | 182 (523) | 89 (386) |
| 1-6 years Proportion having hospital care/full information (%) | 25/25 (100) | 209/259 (80.7) | 833/1233 (67.6) |
| 7-12 years Mean/year € | 1456 | 1180 | 455 |
| 7-12 years Median (IQR)/year € | 673 (1545) | 93 (842) | 44 (280) |
| 7-12 years Proportion having hospital care/full information (%) | 43/48 (89.6) | 243/379 (64.1) | 945/1793 (52.7) |
| 13-20 years Mean/year € | 1917 | 1274 | 530 |
| 13-20 years Median (IQR)/year € | 837 (1412) | 443 (1505) | 94 (520) |
| 13-20 years Proportion having hospital care/full information (%) | 24/26 (92.3) | 109/134 (81.3) | 409/603 (67.8) |

Includes all inpatient and outpatient care for both psychiatric and somatic reasons. Estimated among those with full hospital care information in each age category available. Due to lack of outpatient visit data 1992–1997 and lack of hospital data after 2016, hospital information is complete only for the age group 7–12 years. All three groups differed significantly in each age category with p-value <0.05.

PAE without FASD exceeded the FASD group in psychiatric costs even though the FASD group had a greater prevalence in psychiatric care (Fig 1). As expected, controls had significantly lower health care costs than both PAE groups. Mean cumulative hospital care costs for 0–20 years were 55 548€ for FASD (median 33 678€, IQR 56 799€), 30 057€ for PAE without FASD (median 14 651€, IQR 25 064€) and 15 641€ for controls (median 6340€, IQR 11 952€). Among those with PAE, during the first ten years, the FASD-diagnosed group had significantly higher hospital costs reflecting more somatic conditions such as neonatal problems and anomalies, whereas the PAE without FASD group had higher psychiatric costs, though psychiatric care was common in both PAE groups (Figs 1 and 2).

## Out-of-home care and its costs

Compared to the control group, out-of-home care was prevalent in both PAE groups (Fig 3). In FASD group the risk for out-of-home care was highest, though even in the PAE without FASD group the probability of out-of-home placement was over 50%. However, the cumulative prevalence was markedly different between the three study groups. In PAE without FASD and control groups the distributions remained extremely skewed through all age categories, whereas in PAE with FASD group median costs neared the mean costs with increasing age (Table 4). This was because in the 13–18 years' age category 83.3% of the persons in the PAE with FASD group were placed in out-of-home care. In PAE without FASD and control groups the proportions were 40.6% and 5.3%, respectively.

## Total aggregated hospital and out-of-home care costs

Total aggregated estimated costs differed significantly between both PAE groups and controls and constituted mainly of out-of-home care costs reflecting their patterns (Fig 4). Even though at the group level out-of-home care costs far exceeded hospital costs in PAE groups, individual variation was significant. Costs were skewed, particularly among controls, less than one quarter caused 90% of their total costs. (Fig 5).

Individuals who incurred most costs were significantly overrepresented in PAE groups, as total aggregated hospital and out-of-home care costs/per person exceeded 1 million euros in 9 (18.8%) of the FASD group, 46 (12.1%) of the PAE without FASD group and in 10 (0.6%) of controls (FASD vs controls RR 33.7; 95% CI 14.3–79.0, p<0.0001, PAE without FASD vs controls RR 21.8; 95% CI 11.1–42.8, p<0.0001).

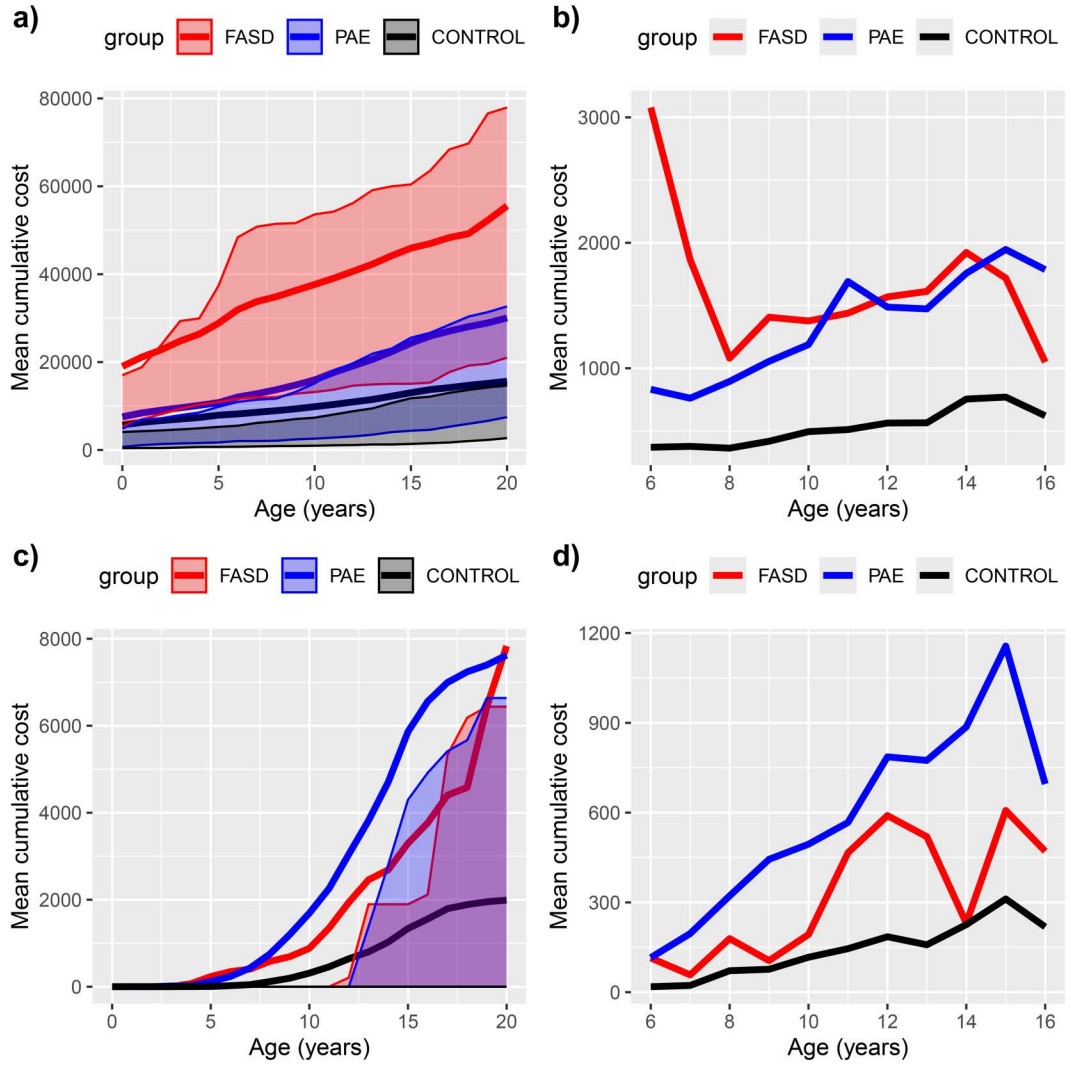

**Fig 2. Inpatient and outpatient hospital care costs (mean and interquartile range, IQR) per individual until the age of 20 years, a) all cumulative costs, b) all annual costs, c) psychiatric cumulative costs, d) annual psychiatric hospital costs.** Due to missing outpatient data before the year 1998 and lack of hospital data after the year 2016, mean annual costs for each age are presented only for ages 6-16 years with complete data **(b, d)**. The mean cumulative costs include imputations for missing data **(a, c)**, whereas annual costs (b, d) do not. Control group is drawn black, whereas PAE with and without FASD diagnosis are drawn red and blue, respectively.

Among the PAE children without a FASD diagnosis was a subgroup (126 children) with a cost distribution similar to the FASD group (Fig 6). The group consisted of the PAE children diagnosed with ICD-10 codes F70 – F98 and history of out-of-home care. The cost distributions between the subgroup of PAE with a FASD profile and the FASD group was not distinguishable (p = 0.945), whereas the comparison to the PAE children without the FASD profile yielded statistically significant difference (p < 0.0001).

## Discussion

Growing up after prenatal alcohol exposure is associated not only with heightened risk for several somatic and psychiatric conditions, as shown also in previous research [7,9,51] but also with significantly higher costs caused by hospital care,

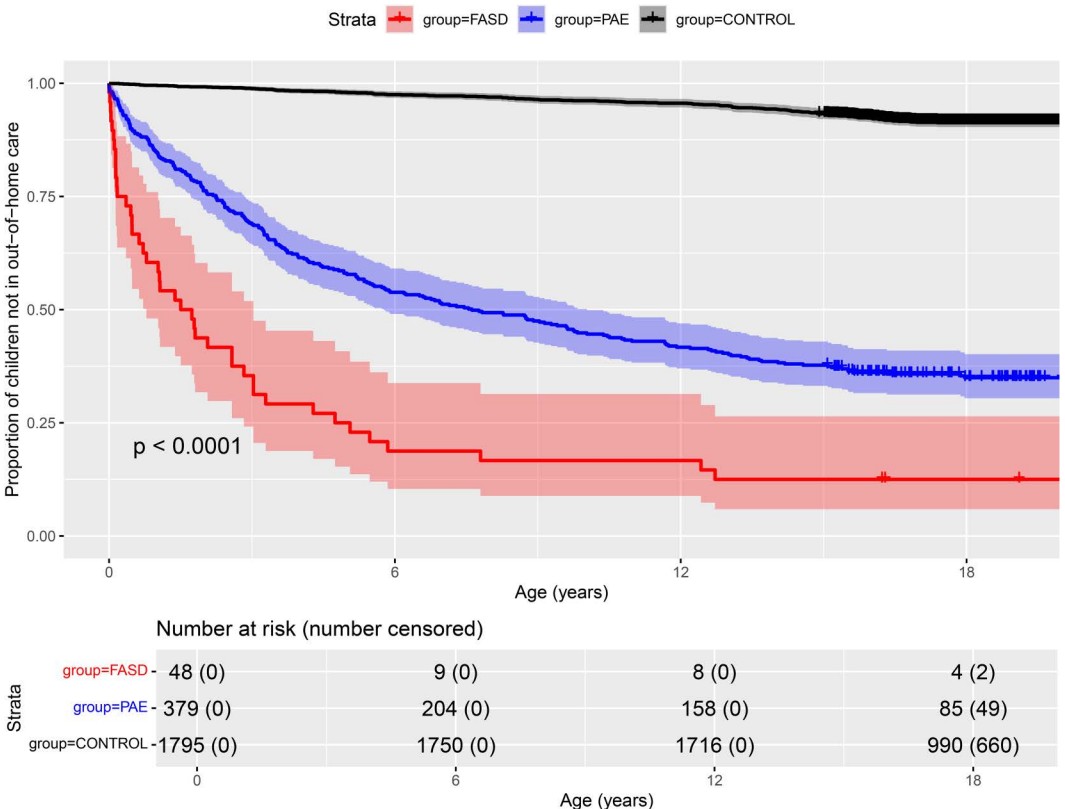

**Fig 3. Kaplan-Meier curve for out-of-home care as proportion of children without any out-of-home care placements, each step down signifies a person placed for the first time out-of-home.**

**Table 4. Out-of-home care costs in each age category, rounded to full euros.**

| Category | FASD | PAE no FASD | Controls |
|---|---|---|---|
| < 1 year Mean € | 4285 | 1964 | 37 |
| < 1 year Median (IQR) € | 0 (0) | 0 (0) | 0 (0) |
| < 1 year Proportion in out-of-home care (%) | 10/48 (20.8%) | 31/379 (8.2%) | 4/1795 (0.2%) |
| 1-6 years Mean/year € | 29996 | 14681 | 538 |
| 1-6 years Median (IQR)/year € | 24917 | 0 (0) | 0 (0) |
| 1-6 years Proportion in out-of-home care (%) | 39/48 (81.2%) | 174/379 (46.2%) | 47/1795 (2.6%) |
| 7-12 years Mean/year € | 35460 | 22766 | 1157 |
| 7-12 years Median (IQR)/year € | 28089 (41999) | 0 (44697) | 0 (0) |
| 7-12 years Proportion in out-of-home care (%) | 39/48 (81.2%) | 187/379 (49.3%) | 61/1795 (3.4%) |
| 13-18 years Mean/year € | 28277 | 14974 | 1296 |
| 13-18 years Median (IQR)/year € | 28144 (26964) | 0 (29386) | 0 (0) |
| 13-18 years Proportion in out-of-home care (%) | 40/48 (83.3%) | 154/379 (40.6%) | 95/1795 (5.3%) |
| Cumulative total costs for 0–19 years, Mean (Median, IQR) | 610 036 (561 304,375 780) | 344 274 (100 449, 621 857) | 20 496 (0, 0) |

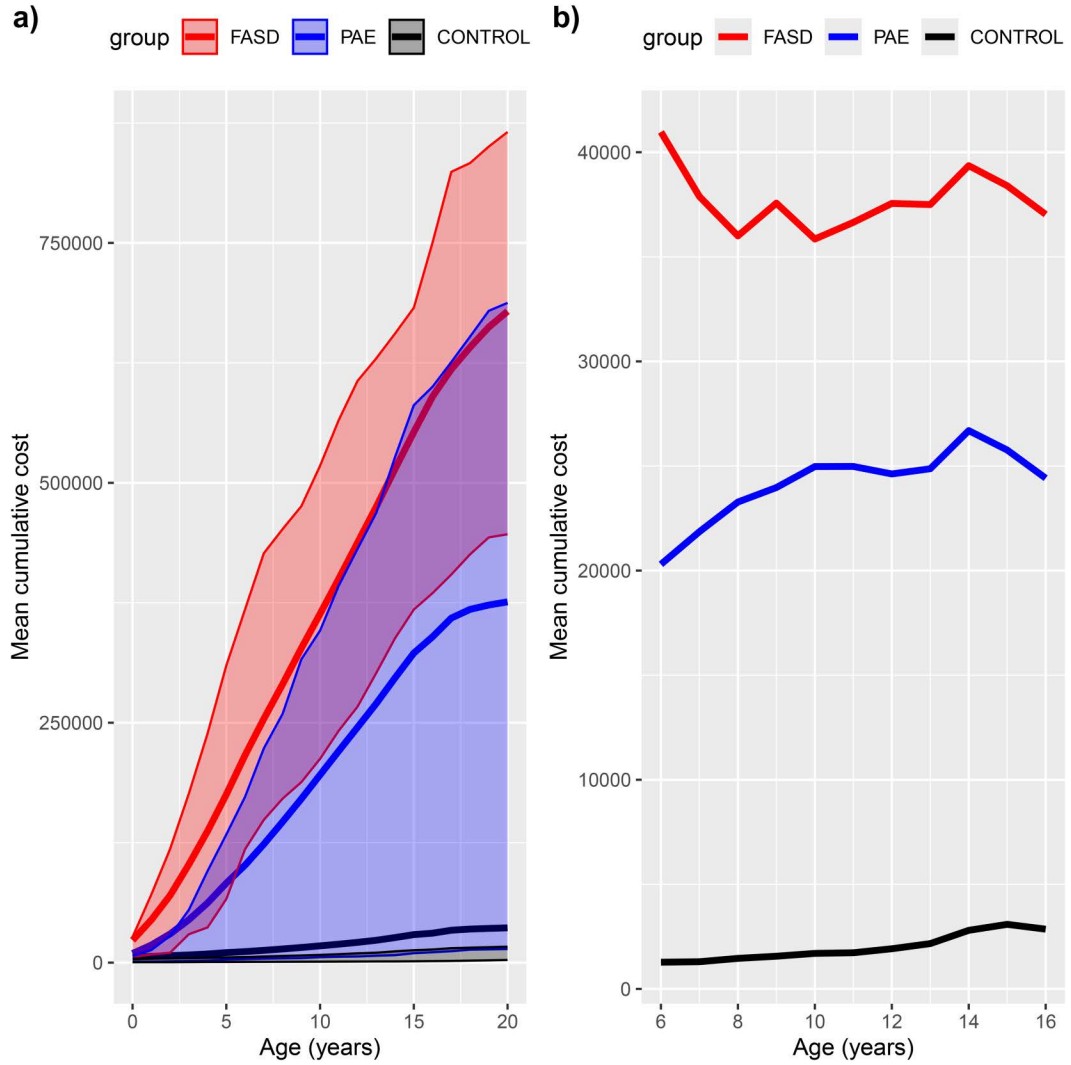

**Fig 4. Total aggregated hospital and out-of-home care costs (red FASD, blue PAE without FASD and black Controls, dotted lines IQR).** Cumulative costs according to the age are shown in subfigure **a)**, whereas the age specific mean annual costs are shown in subfigure **b)**.

and especially by out-of-home care, including foster and residential care, compared to controls. Out-of-home care costs far exceed hospital care costs as in our previous clinical population study [14].

Those diagnosed with FASD had the highest cumulative costs. However, the difference between the two PAE groups diminished with age and psychiatric costs became more prominent over time in those with PAE without FASD diagnosis. Interestingly, among those with PAE, a FASD diagnosis appeared protective against traumatic injuries, substance use disorders, and teenage pregnancies. This effect may reflect increased support, supervision and better meeting their needs following a FASD diagnosis, as caregivers become more aware of risks and needs associated with FASD. Our earlier clinical FASD population study showed that many children diagnosed with FASD had extensive support for school including small group with high adult-pupil-ratio and taxi-transfers between school and home [14], which could be one explanation for reduced risk for injuries. As a FASD diagnosis was associated with early out-of-home placement, the protective effect could be caused by having a more stable foster family placement and

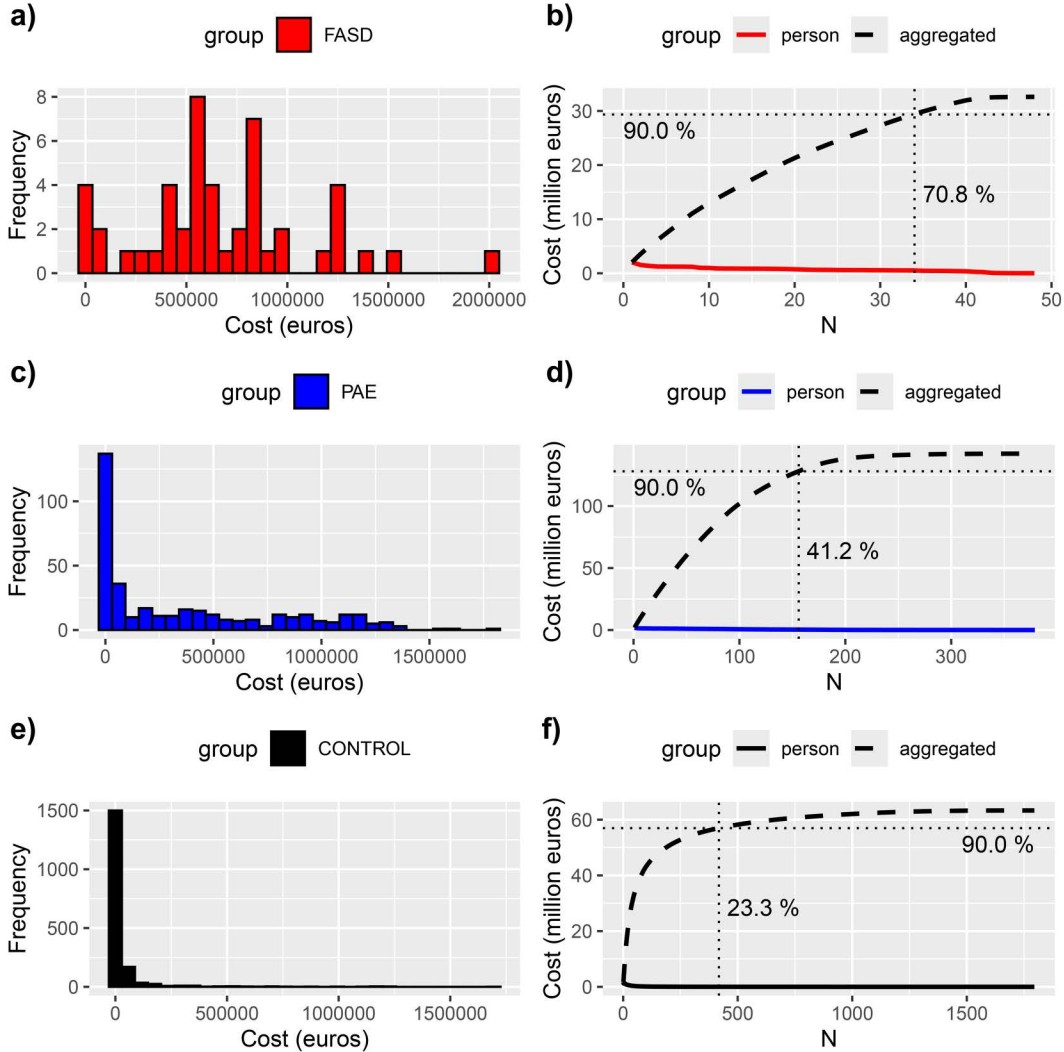

**Fig 5. Aggregated costs of hospital and out-of-home care in the groups, red = FASD, blue = PAE without FASD and black = controls.** On the left panel (subfigures a, c, and **e)**, the histograms present the cost distribution within the respective groups. On the right the panels (subfigures b, d, **f)**, the colored curves display the individuals' total cost, and the dashed curve the cumulative sum of the costs for the respective group. The solid vertical lines mark where the cumulative sum reaches 90% of the aggregated total costs of the group. The further left the vertical line is, the more skewed is the cost distribution. The crossing vertical solid line points to the cumulative cost of the whole respective group.

less to-and-from placements between biological family and out-of-home care. However, statistically early out-of-home care placement did not explain the protective effect of having a FASD diagnosis, supporting earlier findings that an early FASD diagnosis itself may improve outcomes [52]. To our knowledge this is the first study showing that in PAE population having a FASD diagnosis could have protective effect on traumatic injuries. More research on the topic is warranted.

Compared to others with PAE, FASD diagnosis was associated with more maternal self-disclosed daily alcohol use and visible physical effects of PAE, including congenital anomalies, ophthalmologic issues, low birth weight, and these factors likely facilitated a FASD diagnosis. Using the National Register of Congenital Malformations as a source likely increased anomaly detection in all groups compared to studies relying solely on clinical data.

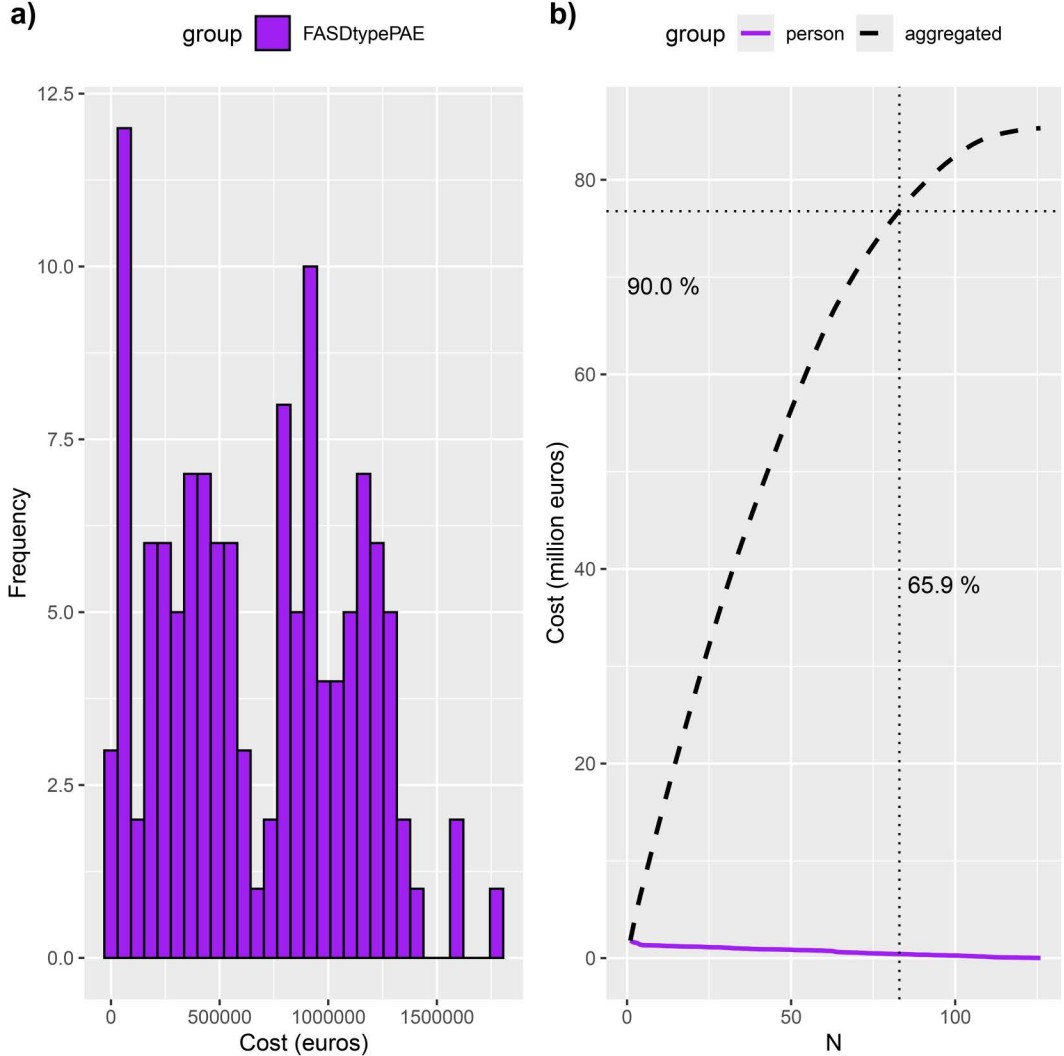

**Fig 6. Aggregated costs of hospital and out-of-home care in the subgroup of PAE children that display similar behavioral problem profile to the FASD children.** In subfigure a, the histogram presents the cost distribution within the subgroup. In subfigures b, the colored curve displays the individuals' total cost, and the dashed curve the cumulative sum of the costs for the subgroup. The solid vertical lines mark where the cumulative sum reaches 90% of the aggregated total costs of the group. The further left the vertical line is, the more skewed is the cost distribution. The crossing vertical solid line points to the cumulative cost of the whole subgroup.

Despite differences in somatic conditions, there were no significant differences between PAE groups in developmental or psychiatric diagnoses such as developmental language disorder, attention deficit hyperactivity disorder, mixed disorders of conduct and emotions, emotional disorders of childhood, depressive episodes, anxiety disorders, stress reactions and adjustment disorders. Children in both PAE groups differed significantly from controls in all these categories. Outcome of PAE can be influenced by the pattern of exposure in addition to timing and dosage of exposure [6]. Weekly alcohol use, which in Finland typically implies binge drinking on weekends, was more common than daily alcohol use in PAE without FASD, which could be one explanation behind more psychiatric than somatic pattern of diagnoses. However, the degree of inexactness and unreliability in self-reporting pattern of alcohol use and limitations of register-based data did not allow detailed analysis for this.

In our data, the costs occurred from 1992 to 2016. Selecting the year 2023 price for reporting costs facilitates the comprehension of their magnitude for a contemporary reader. For time periods without available nominal hospital fee lists, the above-mentioned double index correction is necessary as health and social care costs have not consistently followed consumer price index.

The estimated average total hospital care and out of home care costs until the age of 20 years for a person with FASD was 665 600€ and for a person with PAE without FASD was 374 300€, compared with an average of 36 100€ in controls. Cost analysis indicated that among those with PAE without FASD diagnosis, those in the upper cost quartile resembled FASD diagnosed group suggesting underdiagnosed FASD, also according to their diagnoses, while those in the lower quartile resembled controls, possibly reflecting lower exposure or less severe effects of PAE.

The cost estimations in both PAE groups are naturally lower than total costs described for persons diagnosed with FAS in Sweden [12], despite having comparable social and health care system, or in our Finnish clinical FASD population study [14], as only hospital and out-of-care costs were analyzed in this study and significant proportion of costs in FASD are caused by special schooling arrangements, rehabilitation, disability allowances, other social support services, primary health care, judicial costs and loss on parental income. Compared with other chronic conditions of childhood, health care costs in this study were significantly higher than in asthma [13] and comparable to those of childhood epilepsy [53], however, social care costs were significantly higher in PAE and FASD.

Although special education costs could not be directly estimated using current registers, diagnoses indicating the need for special educational support were available. Based on these, more than half of children with FASD and nearly one third of those with PAE without FASD likely required special school arrangements, compared to less than one in five of controls. However, the actual number of children receiving support is likely higher, given that in Finland school-based services can be provided following an evaluation by a school psychologist, without a formal diagnosis given in hospital.

Similarly, developmental disorders typically requiring early rehabilitation were diagnosed in over one third of those with FASD and in over one in six for others with PAE. These figures also underestimate the true rehabilitation prevalence, as many children receive early intervention (e.g., speech, occupational, or physical therapy) through municipal health services without receiving a formal diagnosis. Only those with severe developmental problems are referred to hospital clinics and receive a developmental diagnosis. Medication costs were not analyzed in this study, though data on relevant diagnoses typically requiring medication, could allow for crude future estimation. Including such unmeasured costs would likely increase total cost estimates. Based on our findings and earlier studies from Sweden and Finland, the true total costs of growing up with FASD may approach or exceed one million euros per individual.

## Strengths

Finland, like all Nordic countries, has national public health care and comprehensive national registers, which enable longitudinal research without participation or attrition bias. The national registers include detailed data on all hospitalizations with main and secondary diagnoses and length of stay, and out-of-home care episodes, allowing for reliable cost estimations.

The study design included all hospital records for both mothers and children, enabling follow-up from the prenatal period to early adulthood. All PAE cases were identified through specialized antenatal clinics for mothers with substance use disorders, with access to maternal self-reported alcohol use and timing of pregnancy recognition. While self-reports may underestimate true alcohol use [54], overreporting during pregnancy is unlikely.

Importantly, out-of-home care data were available for both cases and controls, allowing direct comparison. The use of multiple controls per case reduced the risk of selection bias.

## Limitations and potential confounding factors

This register-based study had several limitations affecting the completeness of cost estimations. Special education arrangements, parental income loss, broader social welfare services, justice system involvement, medications, and

primary health care comprise a significant part of costs in children and youth with FASD [11,12]. These costs could not be included due to data protection regulations or lack of available data. The Register on Primary Health Care was introduced in Finland only in 2011. As a result, the reported total costs are only a part of all costs associated with PAE and represent a conservative estimate.

Individuals in this study were born between 1.1.1992–21.4.2001 and the follow-up period (1992–2016) varied by birth year, with only a portion of participants having complete data through age 20. Outpatient records were only available from 1998, and neonatal diagnoses from 1996 onward, limiting early data for those born in earlier years. Additionally, data on neonatal hospital stay lengths were missing, underestimating costs for the most affected infants, particularly in the FASD group. However, as birth years were similar across groups, underestimation is likely non-differential, except for neonatal care where FASD cases had more severe early health issues.

Those with FASD diagnosis in this study were actually those with dysmorphic FASD (FAS or partial FAS), as ICD-10 lacks specific codes for FASD with only neurobehavioral disorder, who typically receive only diagnoses for symptoms and comorbidities. Consequently, differences between FASD and non-FASD PAE groups reflect characteristics that aid diagnosing FASD, such as visible anomalies, early out-of-home placement, and maternal disclosure of heavy alcohol use. This limits the generalizability of results for FASD group across the full FASD spectrum, as children with non-dysmorphic FASD are included in PAE without FASD group. However, presence of probable alcohol-related neurodevelopmental disorder type of FASD according to IOM 2016 diagnostic criteria for FASD [5] could be argued in those 33.2% of children with PAE who formed a distinct subgroup within the PAE group with cost distribution similar to the FASD group. In addition to diagnoses indicating significant neurodevelopmental, behavioral and emotional disorders, they also had history of out-of-home care.

Nearly all children with PAE were also exposed to maternal smoking, making it difficult to isolate alcohol effects from concurrent effects of tobacco and alcohol exposure. It has been shown in many previous studies that PAE is more common among those who smoke [14,55–58] and smoking exacerbates toxic and teratogenic effects of alcohol [59–61].

Parental substance use may reduce the likelihood of seeking care or diagnosis for children's developmental needs among those living in their birth family, potentially leading to underdiagnosis and cost underestimation.

Finally, lack of alcohol exposure data in controls likely led to inclusion of undiagnosed FASD cases, as the only inclusion criteria for controls was the absence of hospital care for alcohol-related causes in the year prior to birth. Recognition of PAE in health care is low in Finland, as it is based solely on often unreliable maternal self-reporting [54] and not biomarkers during pregnancy. Also, even recognized PAE will not be visible in register data, if the mother is not referred to voluntary follow-up in specialized antenatal clinic. Assuming the European average of 2% of FASD in the general population [1] would mean there are 36 individuals in the control group with potential undiagnosed FASD. Indeed, there were controls with similar profiles to diagnosed FASD cases, suggesting possible underrecognition in this group. Assuming 36 controls with undiagnosed FASD and a similar rate of those with costs of over 1 million euros in that population compared to the FASD group in this study (18.8%), would explain 7/10 of those with highest costs among controls.

## Conclusion

Socioeconomic and health-related problems associated with PAE cause significant societal costs and a burden for health care and child protection services. High out-of-home care costs indicate that early intensive support for families at risk could not only reduce suffering and attachment problems, but also societal costs. Interventions aimed at prevention of PAE and smoking during pregnancy and supporting those at risk could save children and their families from unnecessary suffering and reduce monetary costs.

## Supporting information

**S1 Appendix. Hospital care.** Health care unit costs, most common diagnoses.
(PDF)

**S2 Appendix. Out-of-home care.** Out-of-home care types and costs.
(PDF)

## Acknowledgments

We thank Trevor Yoak for his excellent help in the preparation of the manuscript.

## Author contributions

**Conceptualization:** Mirjami Jolma, Hanna Kahila.

**Data curation:** Mirjami Jolma, Mikko Koivu-Jolma, Taisto Sarkola, Mika Gissler, Niina-Maria Nissinen, Hanna Kahila, Ilona Autti-Rämö, Anne Koponen.

**Formal analysis:** Mirjami Jolma, Mikko Koivu-Jolma.

**Investigation:** Mirjami Jolma.

**Methodology:** Mirjami Jolma, Mikko Koivu-Jolma, Taisto Sarkola, Mika Gissler, Hanna Kahila, Paulus Torkki.

**Project administration:** Mirjami Jolma, Mikko Koivu-Jolma, Hanna Kahila, Anne Koponen.

**Resources:** Mika Gissler.

**Software:** Mikko Koivu-Jolma.

**Supervision:** Anne Sarajuuri, Paulus Torkki, Ilona Autti-Rämö, Anne Koponen.

**Visualization:** Mikko Koivu-Jolma.

**Writing – original draft:** Mirjami Jolma.

**Writing – review & editing:** Mirjami Jolma, Mikko Koivu-Jolma, Taisto Sarkola, Mika Gissler, Niina-Maria Nissinen, Hanna Kahila, Anne Sarajuuri, Paulus Torkki, Ilona Autti-Rämö, Anne Koponen.

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
