## [Decision Letter · Decision Letter 0]

3 Nov 2025

PONE-D-25-42438A case-control register-based study of long-term health and social care costs among children with prenatal alcohol exposurePLOS ONE

Dear Dr. Jolma,

Thank you for submitting your manuscript to PLOS ONE. After careful consideration, we feel that it has merit but does not fully meet PLOS ONE’s publication criteria as it currently stands. Therefore, we invite you to submit a revised version of the manuscript that addresses the points raised during the review process.

Dear Authors, Both reviewers have provided comments that require your attention. I want to encourage you to please address is comment carefully in your resubmission. I look forward to reading your revised paper.==============================

We look forward to receiving your revised manuscript.

Kind regards,

Nafisa M. Jadavji, PhD, MSc, BSc

Academic Editor

PLOS ONE

Journal Requirements:

4. Please remove all personal information, ensure that the data shared are in accordance with participant consent, and re-upload a fully anonymized data set.

Reviewers' comments:

Reviewer's Responses to Questions

**Comments to the Author**

1. Is the manuscript technically sound, and do the data support the conclusions?

Reviewer #1: No

Reviewer #2: Yes

2. Has the statistical analysis been performed appropriately and rigorously? 

Reviewer #1: Yes

Reviewer #2: I Don't Know

3. Have the authors made all data underlying the findings in their manuscript fully available?

Reviewer #1: Yes

Reviewer #2: Yes

4. Is the manuscript presented in an intelligible fashion and written in standard English?

Reviewer #1: Yes

Reviewer #2: Yes

5. Review Comments to the Author

Reviewer #1: while the methodology of the study suggests a retrospective/non-concurrent design with the children being identified based on exposure andfollowed up for outcome the analysis is of a case control design. The measure of association we would look for would be relative risk rather than odds ratio. However the data is interesting and requires a resubmission after considering the suggestions

Reviewer #2: This is a considered and well-written paper demonstrating a high level of expertise on a complex and nuanced subject. The methodology is sound, and the findings are interesting with potential for international impact. There are some minor changes I would recommend before publication:

Terminology Clarification

The term “Out-of-home care” may be problematic from an international perspective, as it could be misinterpreted to mean hospital admission. It appears the authors are referring to children under the care of the state or local authority, including foster care and possibly specialized mother-and-baby units? Clarifying this terminology early on is important to accessibility to a global audience.

International Comparison to Sweden

The inclusion of figures from Sweden in the introduction. It would strengthen the paper to briefly explain why Sweden is a relevant comparator and highlighting key similarities and differences between the two countries in terms of policy, demographics, or care systems for example.

Table 1 – Combined Risk Factors

It is good that Table 1 includes data on pre-pregnancy alcohol use and unplanned pregnancies. If possible, consider adding a combined metric showing cases where both factors were present. This could offer valuable insight into alcohol consumption prior to the recognition of pregnancy. This is an important factor in future prevention of PAE health promotion efforts.

PAE and Control Group Considerations

The paper appropriately acknowledges that the control group may include individuals with prenatal alcohol exposure (PAE), given the limitations of self-reporting. This along with the recognition that some individuals in the PAE group may have undiagnosed FASD is important. It would be good to see the limitations of recording PAE in general but also specific to this collection method in more detail in the discussion.

Subgroup Analysis – FASD-like Characteristics

The discussion refers to a subgroup within the PAE group that exhibits characteristics similar to FASD. It would be beneficial to describe this subgroup in greater detail and, if feasible, include this data in table form to support.

Exposure Patterns and Outcomes

I recommend expanding the discussion to explore whether differences in exposure patterns—such as episodic drinking during the first trimester—might influence outcomes like depression and anxiety. Some evidence suggests that timing and pattern of exposure may be as important as dose.

Cost Comparisons

In the discussion (and the introduction) including a comparison of the total costs associated with FASD and PAE to those of other prevalent conditions would provide useful context and underscore the broader public health implications.

Reduced Injury Rates in FASD Group

The finding of reduced injury rates in the FASD group is intriguing. Are there any known factors specific to Finland that might explain this? A comparison with data from other countries and further exploration in the discussion would be valuable. This could be an area for further research which could be highlighted in the paper too.

6. PLOS authors have the option to publish the peer review history of their article (what does this mean? ). If published, this will include your full peer review and any attached files.

**Do you want your identity to be public for this peer review?** For information about this choice, including consent withdrawal, please see our Privacy Policy .

Reviewer #1: **Yes:** KURYAN GEORGE

Reviewer #2: No

---

## [Author Response · Author response to Decision Letter 1]

6 Dec 2025

Editor: Thank you for submitting your manuscript to PLOS ONE. After careful consideration, we feel that it has merit but does not fully meet PLOS ONE’s publication criteria as it currently stands. Therefore, we invite you to submit a revised version of the manuscript that addresses the points raised during the review process.

Our response: We thank you for the positive feedback. Below we give our response to the specific editorial comments and the comments from the reviewers.

Our response: Files have been named according to the instructions.

Our response: Data availability clause has now been refined: “The data that support the findings of this study are available from the Hospital District of Helsinki and Uusimaa (HUS). However, restrictions apply to the availability of these sensitive data, which were used under license for the current study. Only those researchers who are named in the study permissions have access to the data, and no part of the data can be shared or placed in public repositories. Similar data can be applied from Findata Finnish Social and Health Data Permit Authority Findata https://findata.fi/en/. Link to the current legislation: https://findata.fi/en/services-and-instructions/legislation/.”

Our response: Captions for Supporting Information have been added.

4. Please remove all personal information, ensure that the data shared are in accordance with participant consent, and re-upload a fully anonymized data set.

Our response: Due to the strict research permit, no information that relates to individuals has been uploaded.

Our response: There were no recommendations for previously published works. 

Our response: The reference list has been reviewed.

Reviewer #1: while the methodology of the study suggests a retrospective/non-concurrent design with the children being identified based on exposure and followed up for outcome the analysis is of a case control design. The measure of association we would look for would be relative risk rather than odds ratio. However, the data is interesting and requires a resubmission after considering the suggestions.

Our response: We thank the reviewer for the interest in our study. We have now replaced odds ratios with risk ratios in group comparisons.

Reviewer #2: This is a considered and well-written paper demonstrating a high level of expertise on a complex and nuanced subject. The methodology is sound, and the findings are interesting with potential for international impact. There are some minor changes I would recommend before publication:

Our response: We thank the reviewer for the interest in our study and for several useful suggestions to improve our manuscript. We give our point-by-point responses to the specific suggestion below.

Terminology Clarification

The term “Out-of-home care” may be problematic from an international perspective, as it could be misinterpreted to mean hospital admission. It appears the authors are referring to children under the care of the state or local authority, including foster care and possibly specialized mother-and-baby units? Clarifying this terminology early on is important to accessibility to a global audience.

Our response: Out-of-home care has now been clarified (in abstract line 30) to include placements in foster or residential care. It has also been clarified similarly in Introduction (line 76) and in Discussion (line 357).

International Comparison to Sweden

The inclusion of figures from Sweden in the introduction. It would strengthen the paper to briefly explain why Sweden is a relevant comparator and highlighting key similarities and differences between the two countries in terms of policy, demographics, or care systems for example.

Our response: This is an excellent suggestion, as the economic, societal, historic, demographic, cultural and geographical similarities between Sweden and Finland might not be self-evident to international readers. Brief explanation of similarities and comparability between Sweden and Finland has now been added to Introduction (lines 63-64). The information about Sweden having a comparable social and healthcare system to Finland has also been added to Discussion (line 398).

Table 1 – Combined Risk Factors

It is good that Table 1 includes data on pre-pregnancy alcohol use and unplanned pregnancies. If possible, consider adding a combined metric showing cases where both factors were present. This could offer valuable insight into alcohol consumption prior to the recognition of pregnancy. This is an important factor in future prevention of PAE health promotion efforts.

Our response: The combination of unplanned pregnancy and daily or weekly alcohol use have now been added as additional rows in Table 1.

PAE and Control Group Considerations

The paper appropriately acknowledges that the control group may include individuals with prenatal alcohol exposure (PAE), given the limitations of self-reporting. This along with the recognition that some individuals in the PAE group may have undiagnosed FASD is important. It would be good to see the limitations of recording PAE in general but also specific to this collection method in more detail in the discussion.

Our response: Unrecognition is a very important topic. More discussion about presence of probable undiagnosed FASD in PAE population has been added (lines 452-456). Also, description of problems in recognition and recording of PAE in antenatal care and limitations of register data has been added to limitations (lines 463-472).

Subgroup Analysis – FASD-like Characteristics

The discussion refers to a subgroup within the PAE group that exhibits characteristics similar to FASD. It would be beneficial to describe this subgroup in greater detail and, if feasible, include this data in table form to support.

Our response: This subgroup of PAE exhibiting typical symptomatic diagnoses and cost patterns has now been addressed in the Results section as text (lines 327-331) and the new Figure 6. Its similarity to FASD diagnosed both in cost and diagnose patterns had been added to Discussion (lines 393-396)

Exposure Patterns and Outcomes

I recommend expanding the discussion to explore whether differences in exposure patterns—such as episodic drinking during the first trimester—might influence outcomes like depression and anxiety. Some evidence suggests that timing and pattern of exposure may be as important as dose.

Our response: This is a very interesting and important topic warranting more research. The Discussion has been expanded to include this subject (lines 382-386).

Cost Comparisons

In the discussion (and the introduction) including a comparison of the total costs associated with FASD and PAE to those of other prevalent conditions would provide useful context and underscore the broader public health implications.

Our response: This is another great suggestion for improvement. Comparison with costs of childhood asthma and epilepsy have been added (lines 66-68 and 401-404). Unfortunately, cost analyses for many chronic childhood conditions are still lacking.

Reduced Injury Rates in FASD Group

The finding of reduced injury rates in the FASD group is intriguing. Are there any known factors specific to Finland that might explain this? A comparison with data from other countries and further exploration in the discussion would be valuable. This could be an area for further research which could be highlighted in the paper too.

Our response: This is indeed an intriguing and rather surprising finding that requires further research. Earlier studies have indicated that early diagnosis of FASD has protective effect on many adverse health outcomes, but we could not find other studies researching or showing protective effect on traumatic injuries. Discussion about some possible explanation this association in Finland is added (lines 361-372).

---

## [Decision Letter · Decision Letter 1]

12 Jan 2026

PONE-D-25-42438R1A case-control register-based study of long-term health and social care costs among children with prenatal alcohol exposurePLOS One

Dear Dr. Jolma,

Thank you for submitting your manuscript to PLOS ONE. After careful consideration, we feel that it has merit but does not fully meet PLOS ONE’s publication criteria as it currently stands. Therefore, we invite you to submit a revised version of the manuscript that addresses the points raised during the review process.

We look forward to receiving your revised manuscript.

Kind regards,

Nafisa M. Jadavji, PhD, MSc, BSc

Academic Editor

PLOS One

Journal Requirements:

Additional Editor Comments (if provided):

Reviewer 1 has asked that all references to case control studies be removed, since the study is a non-concurrent cohort study.

Reviewers' comments:

Reviewer's Responses to Questions

**Comments to the Author**

1. If the authors have adequately addressed your comments raised in a previous round of review and you feel that this manuscript is now acceptable for publication, you may indicate that here to bypass the “Comments to the Author” section, enter your conflict of interest statement in the “Confidential to Editor” section, and submit your "Accept" recommendation.

Reviewer #1: All comments have been addressed

Reviewer #2: All comments have been addressed

2. Is the manuscript technically sound, and do the data support the conclusions?

Reviewer #1: Yes

Reviewer #2: Yes

3. Has the statistical analysis been performed appropriately and rigorously? 

Reviewer #1: Yes

Reviewer #2: I Don't Know

4. Have the authors made all data underlying the findings in their manuscript fully available?

Reviewer #1: Yes

Reviewer #2: Yes

5. Is the manuscript presented in an intelligible fashion and written in standard English?

Reviewer #1: Yes

Reviewer #2: Yes

6. Review Comments to the Author

Reviewer #1: PLEASE REMOVE ALL REFERENCES TO CASE CONTROL STUDIES AS THIS IS A NON-CONCURRENT COHORT STUDY. OTHERWISE tHIS MANUSCRIPT IS FINE

Reviewer #2: Overall this is an very interesting piece of research with impactful findings. I can see my comments have been addressed satisfactorily and I am happy to recommend publishing this paper with a few minor amendments.

Please consider rephrasing the line 42 of abstract. "..support and rehabilitation for birthmothers with alcohol consumption during and after pregnancy" (and line 475 of conclusion) to be less blaming and less directed at birth mothers, also the term rehabilitation used in this line heavily suggests all prenatal alcohol exposure results from dependent drinking which would be incorrect.

It may be helpful to phrase this as "prevention of alcohol exposed pregnancies and increased support need" for example.

Line 119 where it reads "mental and behavioural", I think should be "mental health and behavioural".

Kind regards

7. PLOS authors have the option to publish the peer review history of their article (what does this mean? ). If published, this will include your full peer review and any attached files.

**Do you want your identity to be public for this peer review?** For information about this choice, including consent withdrawal, please see our Privacy Policy .

Reviewer #1: No

Reviewer #2: No

---

## [Author Response · Author response to Decision Letter 2]

13 Jan 2026

Response to Editor and Reviewer

Editor:

Thank you for submitting your manuscript to PLOS ONE. After careful consideration, we feel that it has merit but does not fully meet PLOS ONE’s publication criteria as it currently stands. Therefore, we invite you to submit a revised version of the manuscript that addresses the points raised during the review process.

1) Reviewer 1 has asked that all references to case control studies be removed, since the study is a non-concurrent cohort study.

Our response: We thank the editor for the encouragement. Following the reviewers suggestion, we have removed all instances where we refer our study as “case-control”. Additionally, below we give our response to the comments from the reviewers.

Reviewer #1:

1) PLEASE REMOVE ALL REFERENCES TO CASE CONTROL STUDIES AS THIS IS A NON-CONCURRENT COHORT STUDY. OTHERWISE tHIS MANUSCRIPT IS FINE

Our response: We thank the reviewer for the continuing interest in our manuscript and for the time spent reviewing it. We have removed all instances of “case-control” from the title and the text, as requested.

Reviewer #2:

1) Overall this is an very interesting piece of research with impactful findings. I can see my comments have been addressed satisfactorily and I am happy to recommend publishing this paper with a few minor amendments.

Our response: We thank the reviewer for the positive response and for the time spent reviewing the manuscript.

2) Please consider rephrasing the line 42 of abstract. "..support and rehabilitation for birthmothers with alcohol consumption during and after pregnancy" (and line 475 of conclusion) to be less blaming and less directed at birth mothers, also the term rehabilitation used in this line heavily suggests all prenatal alcohol exposure results from dependent drinking which would be incorrect.

It may be helpful to phrase this as "prevention of alcohol exposed pregnancies and increased support need" for example.

Our response: We have rephrased the passages in abstract and conclusions following the suggestion.

3) Line 119 where it reads "mental and behavioural", I think should be "mental health and behavioural".

Our response: The phrase “mental and behavioural disorders” refers to the name of a class of diagnoses in ICD-10 classification.

---

## [Decision Letter · Decision Letter 2]

28 Jan 2026

A register-based study of long-term health and social care costs among children with prenatal alcohol exposure

PONE-D-25-42438R2

Dear Dr. Jolma,

We’re pleased to inform you that your manuscript has been judged scientifically suitable for publication and will be formally accepted for publication once it meets all outstanding technical requirements.

Kind regards,

Nafisa M. Jadavji, PhD, MSc, BSc

Academic Editor

PLOS One

Additional Editor Comments (optional):

Reviewers' comments:

Reviewer's Responses to Questions

**Comments to the Author**

1. If the authors have adequately addressed your comments raised in a previous round of review and you feel that this manuscript is now acceptable for publication, you may indicate that here to bypass the “Comments to the Author” section, enter your conflict of interest statement in the “Confidential to Editor” section, and submit your "Accept" recommendation.

Reviewer #1: All comments have been addressed

2. Is the manuscript technically sound, and do the data support the conclusions?

Reviewer #1: Yes

3. Has the statistical analysis been performed appropriately and rigorously? 

Reviewer #1: Yes

4. Have the authors made all data underlying the findings in their manuscript fully available?

Reviewer #1: Yes

5. Is the manuscript presented in an intelligible fashion and written in standard English?

Reviewer #1: Yes

6. Review Comments to the Author

Reviewer #1: this study deals with an important issue and needs to be highlighted. However the faalcies of using record based information must be highlighted

7. PLOS authors have the option to publish the peer review history of their article (what does this mean? ). If published, this will include your full peer review and any attached files.

**Do you want your identity to be public for this peer review?** For information about this choice, including consent withdrawal, please see our Privacy Policy .

Reviewer #1: No

---

## [Editor Report · Acceptance letter]

PONE-D-25-42438R2

PLOS One

Dear Dr. Jolma,

I'm pleased to inform you that your manuscript has been deemed suitable for publication in PLOS One. Congratulations! Your manuscript is now being handed over to our production team.

Kind regards,

on behalf of

Dr. Nafisa M. Jadavji

Academic Editor

PLOS One